# Dual Efficiency and Productivity Analysis of Renewable Energy Alternatives of OECD Countries

Sedef E. Kara [1], Mustapha D. Ibrahim [2,*] and Sahand Daneshvar [1]

1   Department of Industrial Engineering, Faculty of Engineering, Eastern Mediterranean University, Via Mersin 10, Famagusta 99628, TRNC, Turkey; esmasedefkara@gmail.com (S.E.K.); sahand.daneshvar@emu.edu.tr (S.D.)
2   Industrial Engineering Technology, Engineering Technology & Science Faculty of Engineering, Higher Colleges of Technology, Sharjah P.O. Box 7947, United Arab Emirates
*   Correspondence: mibrahim1@hct.ac.ae; Tel.: +971-554-935318

**Abstract:** This paper examines the dual efficiency of bioenergy, renewable hydro energy, solar energy, wind energy, and geothermal energy for selected OECD countries through an integrated model with energy, economic, environmental, and social dimensions. Two questions are explored: Which renewable energy alternative is more dual efficient and productive? Which renewable energy alternative is best for a particular country? Data envelopment analysis (DEA) is used for the efficiency evaluation, and the global Malmquist productivity index is applied for productivity analysis. Results indicate bioenergy as the most efficient renewable energy alternative with a 20% increase in average efficiency in 2016 compared to 2012. Renewable hydro energy, wind energy, and solar energy show a 17.5%, 16%, and 11% increase, respectively. The average efficiency growth across all renewable energy alternatives signifies major advancement. Country performance in renewable energy is non-monolithic; therefore, they should customize their renewable energy portfolio accordingly to their strengths to enhance renewable energy efficiency. Renewable hydro appears to have the most positive productivity change in 2016 compared to 2012, while solar energy regressed in productivity due to its scale inefficiency. All renewable energy alternatives have relatively equal average pure efficiency change. The positive trend in efficiency and productivity provides an incentive for policy makers to pursue further development of renewable energy technologies with a focus on improving scale efficiency.

**Keywords:** efficiency; productivity; renewable energy; data envelopment analysis; Malmquist productivity index; OECD

## 1. Introduction

When countries want to achieve sustainability, they consider renewable energy (RE) as a fundamental driver of sustainable and socio-economic development [1]. There are multiple RE alternatives, and choosing the right one for the country is also critical because investment and resource availability also plays an important role. Climate change as a result of fossil fuels has exacerbated the need for RE as a component of the world energy consumption portfolio [2].

To achieve a sustainable ecosystem, RE has been highlighted by researchers as a path to environmental sustainability. According to a forecast of the International Energy Agency (IEA), the proportion of renewables in primary energy use will rise from 13% in 2011 to 18% in 2035. This will increase the share of RE in the energy mix [3]. RE is ranked second in terms of electricity production owing to the growth of hydropower and bioenergy [3]. Institutions and the government are ramping support for RE technology in order to reduce the cost of production. A number of studies argue that RE reduces greenhouse gas emissions and efficiently utilizes resources better than fossil fuel [4,5].

However, many still argue about the economic implications of RE while producing the needed power.

The environmental and economic concerns of RE are yet to be fully reconciled [6,7]. Decisions that incorporate investors' desire for shareholder maximization and society's concerns for the environment are likely to be sustainable [8]. To incorporate environmental and economic sustainability of wind energy, Welch and Venkateswaran [7] coined the term dual sustainability (efficiency) to mean the achievement of environmental and financial sustainability simultaneously. A similar study is yet to be made for RE alternatives concurrently. A relative comparison among the RE alternatives presents an opportunity for an informed choice of RE. Efficiency is a universal measurement that can be applied to energy alternatives, in this case, RE. The efficiency context and grounds for comparison have to be consistent across all RE alternatives. An array of indicators to define the situation and objective of the analysis needs to be made [9]. In this study, the dual efficiency context and grounds for comparison were consistent across all RE alternatives. Therefore, relative comparison of RE alternatives in different periods across various countries was feasible. This study aimed to fill the void in RE literature by analyzing the dual relative efficiency and productivity in addition to electricity generation of RE alternatives. Benchmark for the RE alternatives and strategic policy recommendation for future investment into RE systems for individual countries is needed to enhance efficiency. To estimate efficiency and productivity of the RE alternatives, this study used data envelopment analysis (DEA) models and Malmquist productivity index that are widely used in applied energy literature [10,11]. The major types of RE sources are: hydropower, biomass, geothermal, ocean, solar, and wind. [12,13].

Hydropower is the most mature and largest source of RE for producing electricity [14]. Hydropower plants produce zero carbon emissions as they convert the energy in flowing water into electricity [15]. Biomass is one of the RE sources capable of making a significant contribution to the world's future energy supply [16]. Bioenergy is the energy derived from biomass (organic matter) such as plants and wastes [17]. Some utilities and power generating companies with coal power plants have found that replacing some coal with biomass is a low-cost option to reduce emissions [18]. In addition, using biomass in boilers reduces nitrous oxide emissions [19]. The most common biofuel is ethanol. Another biofuel is biodiesel, which can be made from vegetable and animal fats. Biodiesel can be used to fuel vehicles or as a fuel additive to reduce emissions [20]. Geothermal energy is the natural heat within the earth that arises from the earth's core. To produce power from geothermal energy, wells are dug a mile deep into underground reservoirs to access the steam and hot water, which can then be used to drive turbines connected to electricity generators [21]. It has strong potential for continued expansion, especially in developing countries. The ocean can produce two types of energy: thermal energy from the sun's heat and mechanical energy from the tides and waves [22]. Electricity conversion systems can use either the warm surface water or boil the seawater to turn a turbine, which activates a generator [23]. Solar energy is the energy that comes from the sun. The energy is used by solar cells which convert sunlight into direct current electricity [22]. The sun is a major source of inexhaustible free energy (i.e., solar energy). Currently, new technologies are being employed to generate electricity from harvested solar energy [24]. Wind turbines transform wind energy into electric energy without producing any waste. Wind energy is a clean source of energy, and wind power is one of the lowest-priced RE technologies available [25]. In recent years, electricity generation from wind energy has grown all over the world.

*Efficiency of Renewable Energy*

From an economic perspective, efficiency is the ratio of resources consumed to the results achieved or the ratio of input to output [26]. Efficiency analysis has grown in complexity because it should not only include economic perspective but also environmental and social dimensions [27]. Results of efficiency analysis have been providing guidance for

policy makers at economic and micro/macro levels. This has helped policy makers make informed decisions suited for their available resources and societal constraints [28]. Financial or economic efficiency is a general concept of efficiency where the desired outcome of the system is compared to the investment made in the particular system [29]. Environmental efficiency, on the other hand, takes into consideration environmental impact of resources consumed in the society [30]. Combining economic and environmental perspectives for efficiency creates a dual efficiency from an energy perspective with energy dimension as the principle factor.

Energy–environmental efficiency is a concept related to environmental consequence of a system toward increase in the desired output (e.g., access to clean energy) and reduction in undesirable environmental output (e.g., carbon emission) through sustainable practices represented by their environmental performance. Environmental efficiency has been considered as an important issue [31]. Edmonds and Reilly [32] argued that decision makers require global environmental efficiency analysis for environmental development as well as energy development in order to model or forecast energy changes for the future.

Extensive research has been carried out on renewable energy [33–35]. Chien, and Hu [36] compared macroeconomic efficiency of OECD and non-OECD in terms of RE. They showed that increasing RE use improves efficiency, while an increase in traditional energy decreases environmental efficiency. Studies such as Ibrahim and Alola [37] also support the findings of Chien and Hu [36]. Efficiency of renewable energy investment for European countries was performed by Cicea et al. [28] with RE supply and generation as critical factors. Energy sustainability is a multifaceted system with economic dimension, environmental dimension, social dimension, and the primary energy dimension. Numerous studies have used different indicators to account for the efficiency of the various energy sustainability dimensions. Labor, capital stock, GDP, and carbon emission were used by Zaim and Taskin [38] to evaluate environmental efficiency and carbon dioxide emissions in the OECD. Cicea et al. [28] used energy intensity, GDP per capita, and GDP per RE investment as inputs and carbon emission as output to analyze environmental efficiency of investment in RE. In energy efficiency, environmental efficiency, and economic efficiency of RE literature, indicators such as labor, net stock of fixed capital, materials, capacity, carbon emission, and renewable electricity generation have been used by numerous studies [36,39–42]. For a comprehensive review of energy efficiency evaluation using DEA, see Xu et al. [43].

Investment made in transforming the RE sources into usable energy comes into consideration when trying to estimate dual efficiency of the RE alternatives. The cheapest energy source might not necessarily be the most efficient or productive when you take into consideration the various stages of energy transformation including, generation, storage, and transportation, in addition to energy sustainability dimensions such as economic, environmental, and social dimension with electricity production as a principal output. This study examined the dual efficiency of RE alternatives considering energy sustainability dimensions. To achieve the said objectives, the remainder of the article is organized as follows: Section 2 discusses the efficiency evaluation method and data sets. Section 3 presents the results and discussion of the analysis. Conclusion and recommendations are made in Section 4.

## 2. Materials and Methods

### 2.1. Inputs and Outputs Factors

Articles in the literature for RE efficiency analysis are void of at least one of the important dimensions of energy sustainability, or the environmental indicator used to represent the environmental dimension lacks robustness in its representation as required by the environmental sustainable development goals (SDGs) target "SDG13, SDG14, and SDG15" [22,44]. In this study, all energy sustainability dimensions were represented in addition to a robust composite indicator for environmental dimension. For economic dimension, capital investment in each RE was considered. Energy dimension, environmental

dimension, and social dimension were considered as outputs, while economic/financial resources were considered as input. Data for the analysis were sourced from the International Renewable Energy Agency [45], World Bank [46], and Yale Center for Environmental Law and Policy [47].

Input

- Economic Dimension: Capital investment (USD billions)—investment in each RE source is considered as input into the technology. Capital investment includes all forms of financial support such as credit line, equity investment, grants, and guarantee toward RE transition [45]. Investment is made in expanding installed capacity and technologies required for RE to usable forms. Investment data are presented in billions of United States dollars (USD billions) at 2017 prices.

Output

- Energy Dimension: Electricity generation from respective RE sources (GWh). This represents the amount of electricity generated from the respective RE alternative [45].
- Environmental Dimension: Environmental Performance Index (EPI). EPI is a data-driven summary of the state of sustainability of a country. It is developed using 32 performance indicators across 11 issue categories under two major issues: environmental health and ecosystem vitality. Figure 1 presents the composition of EPI which makes it a comprehensive indicator for environmental dimension [47]. The EPI offers a powerful policy tool in support of efforts to meet the targets of the UN SDGs and to move society toward a sustainable future. The EPI score indicates which country is best addressing the environmental challenges that face every nation while conducting their economic and infrastructural developments. This indicator helps understand environmental progress and refine policy recommendations [47].

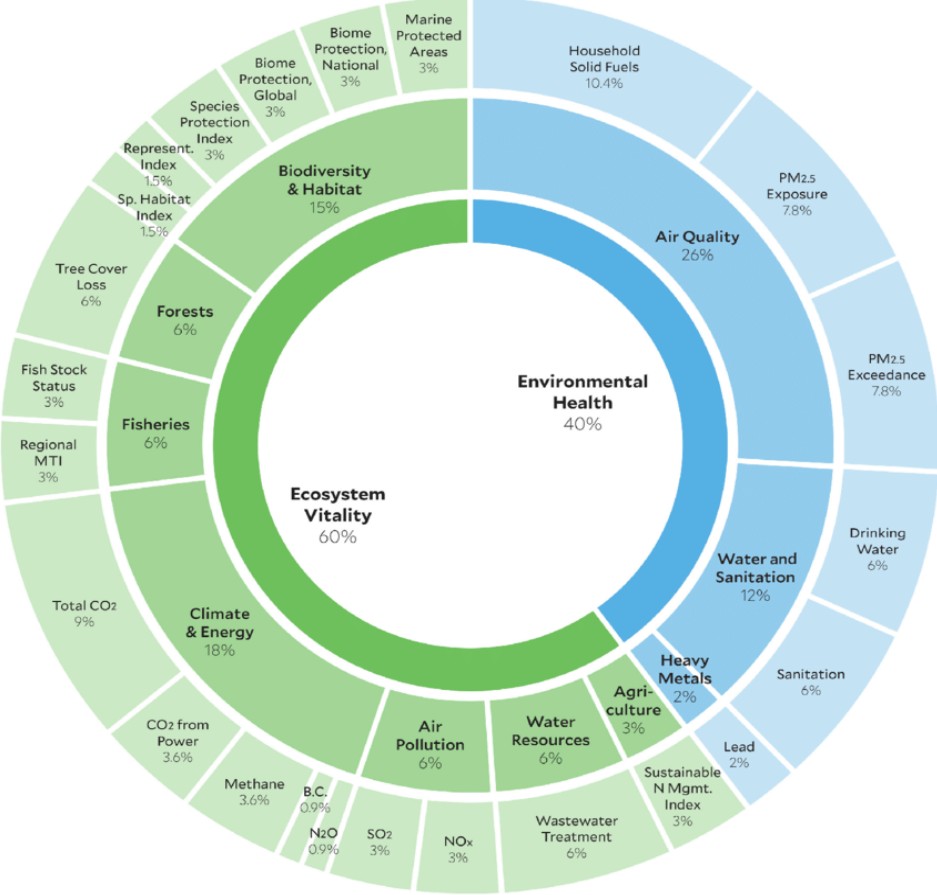

**Figure 1.** Composition of Environmental Performance Index.

- Social Dimension: Access to clean fuels and technologies for cooking—this represents the proportion of the population with access to clean fuels and technology for cooking and domestic activities excluding kerosene [48]. This is a social component of RE to support everyday human activities and a major SDG.

### 2.2. Data Envelopment Analysis

To analyze the dual efficiency of RE alternatives, DEA technique was employed to accommodate the multi-dimension of RE system. DEA is an increasingly popular management tool. DEA was introduced by Charnes, Cooper, and Rhodes through the CCR model [49] and was modified by Banker, Charnes, and Cooper through the BCC model [29]. It measures efficiency of homogenous systems known as decision-making units (DMUs) using frontier estimation. DEA allows for the total factor efficiency of DMUs with multiple inputs and outputs comprising measurement units that cannot be reduced to a common denominator criterion [50]. DEA has grown in popularity in efficiency evaluation of both public and private sectors. In DEA, there are a number of producers (DMUs). In this study, RE systems for each country at a particular year. Each producer takes a set of inputs (Investment) and produces a set of outputs (electricity, environmental performance, and access to clean fuels and technologies). The systems take varying levels of inputs and provide different levels of outputs. DEA attempts to determine which system is most efficient. The fundamental assumption of DEA is that, if a system "A" produces "Y(A)" amount of output with "X(A)" amount of inputs, then other systems should be able to do the same. In the context of dual RE efficiency, if a particular RE utilizes a certain amount of investment, then the output should be compared with other RE alternatives since they are all receiving investment for their development. If a particular RE alternative has better combination of outputs while receiving less investment, then it is considered to be more efficient than others. To illustrate DEA frontier analysis technique, Figure 2 presents a numerical illustration for one input–one output production possibility set of systems for simplicity. Each unit utilizes $x$ amount of input and produces $y$ amount of outputs; DMU($x, y$). DMUs on the frontier are considered to be relatively efficient, and those enveloped are deemed inefficient. A(4, 6), B(6, 9), C(6, 15), D(8, 9), E(10, 21), F(10, 18), and G(12, 15).

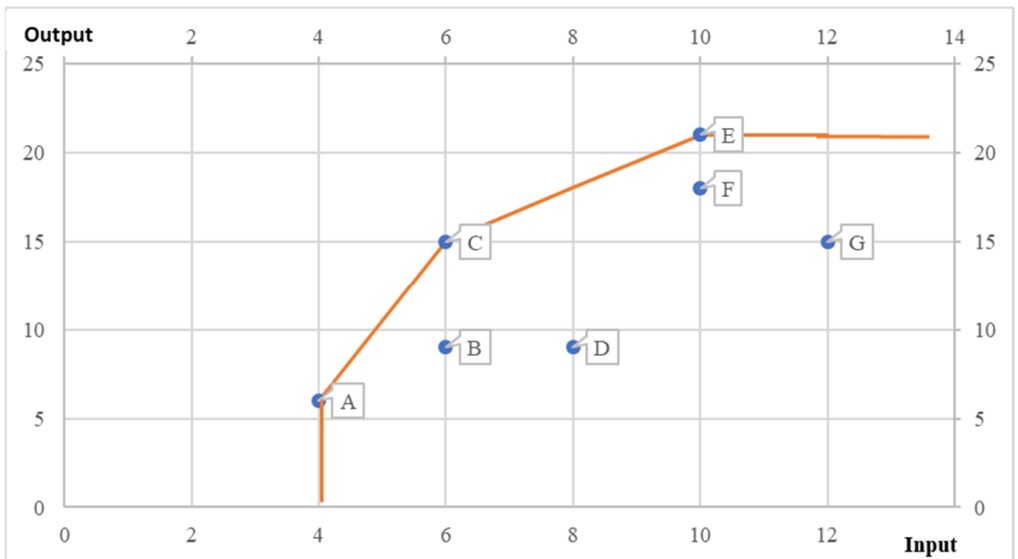

**Figure 2.** Data envelopment analysis efficiency frontier.

Advantages of its application are as follows: it easily accommodates production systems with multiple inputs and outputs, it imposes no functional form for the production function and no endogeneity bias of traditional regression technique, and more importantly,

it identifies improvement targets for the inefficient units to achieve efficiency, thus providing useful insight into sources of inefficiency [51–53]. From this perspective, the DEA approach is valuable for policy makers and management to understand their processes and identify if they are utilizing their resources appropriately by comparing them to the best practice [51]. DEA compares the homogenous units among themselves and accepts the best observation as the efficient frontier, and other observations are benchmarked compared to the frontier. DEA identifies the frontier by seeking the maximum input/output combination. The relative efficiency scores are reported between 0 and 1. DMUs with score of 1 are regarded as efficient relative to other units. Whereas less than 1 is regarded as inefficient. DEA models comprise constant returns to scale (CRS) and variable returns to scale (VRS) models. The CRS model systems assume that increase in inputs results in proportional increase in output level, while VRS models assume that increase in input does not necessarily results in proportional increase in output. The VRS models show if a particular system is evaluated at increasing returns to scale (IRS), constant returns to scale (CRS), or decreasing returns to scale (DRS) [54]. A system is said to operate at IRS if a proportionate increase in its inputs results in more than proportionate increase in output. Conversely, DRS unit results in less than proportionate increase in output [54,55].

Application of DEA in energy efficiency literature is well documented, with some linking it to environmental efficiency. However, only a few examine RE and make comparison with other countries. Chien and Hu [36] compared OECD and non-OECD countries' RE and traditional energy with focus on macro-economic efficiency. Efficiency analysis of renewable energy investment in European countries was performed by Cicea and Marinescu using DEA [28]. This study evaluated the dual efficiency of RE among OECD countries. The analysis has two folds: first, comparison between RE alternatives and then across different countries. Two important conclusions can be made. First, the overall performance of each RE alternative can be estimated. Second, countries can reflect and see which RE alternative is most efficient for them since the comparison is made on homogenous grounds, i.e., investment, which covers installed capacity and other energy transformation process, electricity generation from the RE alternatives, and environmental performance of the country.

To develop the model, consider a set of $n$ observed DMUs; each $DMU_j$, $j = (1, \ldots n)$, utilizes $m$ inputs $x_{ij} = (x_{1j}, \ldots, x_{mj}) > 0$ to produce $s$ outputs $y_{rj} = (y_{1j}, \ldots y_{sj}) > 0$. The DMU represents the RE alternative in a country. A country can utilize multiple RE alternatives simultaneously, for example, Mexico—Bioenergy, Mexico—Solar energy, Finland—Wind energy, and Finland—Hydropower. We assume that all entries of these two arrays are positive. Overall, $n$ DMUs and the production possibility set (*PPS*) are as follows:

$$PPS = \{(x, y) \in \mathbb{R}_+^{m+s} : x \text{ can produce } y\}$$

Model (1) shows the output-oriented BCC VRS DEA model. The dual form that shows input weight $v_i$ and output weight $u_r$ is presented in Model (2). An efficiency score of one indicates the unit as efficient and less than one as inefficient.

$$
\begin{aligned}
&Min \; \varphi \\
&Subject \; to \\
&\sum_{i=1}^{n} z_j x_j - s^+ = x_0, \; i = 1, \ldots, n \\
&\sum_{i=1}^{n} z_j y_j + s^+ = \varphi y_0, \; j = 1, \ldots, n \\
&\sum_{i=1}^{n} z_j = 1 \\
&z_0 \geq 0, \; j = 1, \ldots, n
\end{aligned}
\tag{1}
$$

$$
\begin{aligned}
&Max \sum_{r=1}^{s} u_r y_0 + \sigma \\
&Subject\ to \\
&-\sum_{i=1}^{m} v_i x_{ij} + \sum_{r=1}^{s} u_r y_{rj} + \sigma \le 0 \\
&\sum_{i=1}^{m} v_i x_{ij} = 1 \\
&v_i \ge 0,\ u_r \ge 0,\ \sigma : unrestricted
\end{aligned}
\tag{2}
$$

### 2.3. Malmquist Productivity Index

DEA models analyze the relative efficiency of units; however, the Malmquist productivity index (MPI) used to estimate total factor productivity change (TFPC) examines the change in efficiency between period $t$ and $t + 1$ [56]. MPI is a broadly used method to track progress of system performance in various sectors. Ibrahim et al. [57] applied MPI for healthcare systems, while Sueyoshi and Goto [58] utilized MPI for environmental efficiency of industrialized countries. Similarly, Woo et al. [41] analyzed environmental efficiency of agricultural sector of European countries using MPI [59]. MPI Equation (3) refers to the ratio of the distance functions to measure their productivity [60]. The distance function was extended to DEA-based MPI by Färe and Grosskopf [61] using geometric mean index. MPI can be decomposed into technical efficiency change (TEC) or efficiency change (EC) (Equation (4)) and frontier change (FC) or technical change (TC) as illustrated by Equation (5) [62].

$$
M_t^{t+1} = \left[ \frac{D_0^t\left(x_0^{t+1}, y_0^{t+1}\right) D_0^{t+1}\left(x_0^{t+1}, y_0^{t+1}\right)}{D_0^t\left(x_0^t, y_0^t\right) D_0^{t+1}\left(x_0^t, y_0^t\right)} \right]^{\frac{1}{2}}
\tag{3}
$$

$$
TEC = \frac{D_0^{t+1}\left(x_0^{t+1}, y_0^{t+1}\right)}{D_0^t\left(x_0^t, y_0^t\right)}
\tag{4}
$$

$$
FC = \left[ \frac{D_0^t\left(x_0^{t+1}, y_0^{t+1}\right) D_0^t\left(x_0^t, y_0^t\right)}{D_0^{t+1}\left(x_0^{t+1}, y_0^{t+1}\right) D_0^{t+1}\left(x_0^t, y_0^t\right)} \right]^{\frac{1}{2}}
\tag{5}
$$

From Equation (3), efficiency improves if $M_t^{t+1} > 1$, remains the same if $M_t^{t+1} = 1$, and decreases if $M_t^{t+1} < 1$. Equation (4) estimates the "catch up" effect of the DMU. It measures whether the DMU is closer or further away from the frontier in period $t$ and $t + 1$. FC or TC symbolizes technological progress or regression of the DMU between $t$ and $t + 1$. To overcome possible infeasibility in DEA model and lack of circularity, Pastor and Lovell [63] proposed the global Malmquist index. The output distance indices are measured with respect to a global benchmark technology, defined as the convex hull of the set of all period's technologies. TEC or EC can be further decomposed into pure efficiency change (PEC) and scale efficiency change (SEC). To perform a comprehensive analysis of the RE alternatives. This study employed VRS DEA model and global MPI.

## 3. Results

This study applied the output-oriented BCC model to examine the dual efficiency of RE alternatives for selected OECD countries. Data availability of RE was the only metric used to select the evaluated countries. To ensure empirical stability of DEA models, the number of evaluated units "$n$" must satisfy the criteria: $n \ge \max\{m \times s,\ 3(m + s)\}$ [64]. Considering the one input–three outputs, the number of units should be greater than or equal to 12. The study evaluated sixty-seven units, and therefore the model is stable, and results are reliable. The Performance Improvement Management (PIM-DEA) tool was used for the efficiency analysis. Malmquist productivity index was also applied to examine

the change in efficiency between periods. The RE alternatives evaluated were bioenergy, renewable hydro, solar energy, wind energy, and geothermal energy for the available OECD countries. Renewable hydro and geothermal energy were excluded for 2014 and 2016, respectively, due to insufficient data at those periods. This does not affect the stability of the model or reliability of the results as the PPS function remains the same. The results of the analysis have two folds: the most efficient RE alternative across the evaluated OECD countries and the most efficient RE for an individual country. The second part of the result can infer resource availability of a particular RE alternative.

Figure 3 presents the average efficiency for the evaluated RE alternatives, and Table 1 shows the individual efficiency scores. Average efficiency appears to increase for all RE alternatives across the evaluated period. Bioenergy shows a 20% efficiency increase in 2016 compared to 2012. Renewable hydro shows a 17.5% increase while wind energy shows a 16% increase in efficiency, and solar energy shows an 11.4% increase in 2016 compared to 2012. Reliable data for geothermal energy were available for only three countries, Chile, Mexico, and Turkey 2014, with an efficiency score of 77.9%, 72.8%, and 86.4%, respectively.

Across all RE alternatives in the evaluated period and countries, bioenergy appears to be the most efficient with an average efficiency of 99.3% in 2016, followed by renewable hydro in 2016 with an average efficiency of 96.45%. Wind energy and solar energy have an annual average maximum efficiency of 92.98% in 2016. The continued increase in average efficiency of all RE alternatives can be attributed to the growing technological advancement over the years; however, bioenergy appears to be the most significantly improved form of RE alternative.

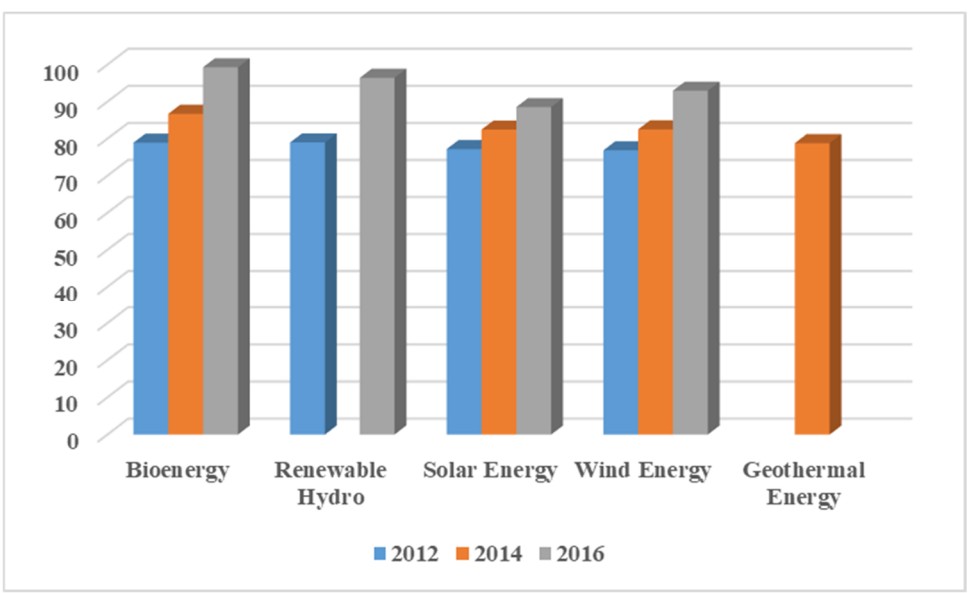

**Figure 3.** Average renewable energy dual efficiency.

Different countries at different periods appear to be the benchmark for individual RE alternatives. Countries that aim to improve certain RE sources can reference the said benchmark countries. Chile and Finland 2016 are the benchmark for bioenergy; France, Italy, and Turkey in 2016 are the benchmark for renewable hydro; Italy in 2016 is the benchmark for solar energy; the USA in 2012 and Sweden in 2016 are the benchmark for wind energy. For geothermal energy, a reference can be made to Turkey in 2014. However, Turkey should focus more on renewable hydro as its primary source of RE. All other RE sources appear to be less efficient for Turkey. The UK's wind energy efficiency appears to be higher than bioenergy in 2012; however, bioenergy has a higher efficiency in 2014 and 2016. Therefore, the UK should focus on enhancing bioenergy systems to improve RE efficiency. Consideration can be given to an RE mix of bioenergy and wind energy for the UK. Similarly, Sweden should focus on an RE mix of bioenergy and wind energy. Countries

can draw a conclusion on which RE alternative to focus on if they are to enhance their RE efficiency and boost their sustainable energy portfolio.

**Table 1.** Dual efficiency score for renewable energy for selected OECD countries.

| 2012 | | | 2014 | | | 2016 | | |
|---|---|---|---|---|---|---|---|---|
| **Energy** | **Countries** | **Effi.** | **Energy** | **Countries** | **Effi.** | **Energy** | **Countries** | **Effi.** |
| Bioenergy | Finland | 71.27 | Bioenergy | Chile | 94.99 | Bioenergy | Austria | 97.41 |
| | Mexico | 93.89 | | France | 82.22 | | Chile | 100 |
| | Spain | 71.57 | | Italy | 88.18 | | Finland | 100 |
| | Sweden | 76.16 | | Mexico | 81.51 | | Italy | 98.59 |
| | UK | 81.77 | | Sweden | 86.34 | | Sweden | 99.96 |
| Renewable Hydro | Austria | 79.26 | | Turkey | 84.38 | | UK | 99.8 |
| | Colombia | 87.43 | | UK | 89.49 | Renewable Hydro | Colombia | 85.78 |
| | Denmark | 70.29 | Geothermal Energy | Chile | 77.9 | | France | 100 |
| Solar | Chile | 61.59 | | Mexico | 72.08 | | Italy | 100 |
| | Colombia | 88.53 | | Turkey | 86.44 | | Turkey | 100 |
| | Mexico | 90.17 | Solar | Chile | 77.66 | Solar | Chile | 86.24 |
| | Spain | 68.53 | | Israel | 72.73 | | Colombia | 85.5 |
| Wind Energy | Austria | 76.24 | | Mexico | 100 | | Italy | 98.48 |
| | Belgium | 69.82 | | Turkey | 79.48 | | Mexico | 92.01 |
| | Denmark | 70.32 | | Austria | 91.08 | | Turkey | 80.52 |
| | Germany | 79.8 | Wind Energy | Chile | 77.67 | Wind Energy | Austria | 95.76 |
| | Ireland | 64.77 | | Finland | 83.72 | | Belgium | 88.68 |
| | Poland | 72.7 | | Germany | 93.89 | | Netherlands | 91 |
| | Spain | 76.07 | | Mexico | 70.39 | | Poland | 93.01 |
| | UK | 82.11 | | Netherlands | 86.28 | | Sweden | 100 |
| | USA | 100 | | Poland | 78.47 | | Turkey | 83.29 |
| | | | | Sweden | 86.39 | | UK | 99.05 |
| | | | | Turkey | 68.95 | | | |
| | | | | UK | 88.5 | | | |

DEA allows for weight flexibility and allocates the appropriate weights for the decision variables when calculating efficiency. The weight distribution highlights the variables that are most significant to efficiency attainment of the unit under evaluation. The average weight distribution (capital investment = 3.67, electricity generation from respective RE sources = 0.056, EPI = 1.13, access to clean fuels and technologies = 0.421) shows that capital investment is the most significant indicator. In the output, environmental performance and access to clean fuels and technology are significant factors for efficiency. Interestingly, all RE alternatives across the evaluated period indicate a DRS performance with the exception of Turkey's renewable hydro energy in 2016. This observation, in addition to the economic dimension identified as the most significant indicator, infers that prudent economic policies toward RE and strategic investment are required to improve efficiency. Energy system combines capital and energy to provide energy service [65]; however, DRS is not unique in energy and electricity sector [66], mostly due to the fact that they are capital-intensive and require frequent maintenance to provide constant and reliable service. Therefore, countries should focus on investing in the right RE for them.

Figure 4 illustrates the average global Malmquist indices for the RE alternatives for 2012–2016, and Table 2 presents the decomposition of the global TFPC for countries with a continuous data set. The TFPC is decomposed into two components: TC and EC. EC is further decomposed into SEC and PEC. Solar energy suffers from significant scale inefficiency. Renewable hydro appears to have the most improved TC in 2016 compared to 2012. The improvement of renewable hydro is consistent across all other productivity indices. The relatively high TC for renewable hydro and wind energy implies that they are

the most technologically advanced RE alternative. Solar energy and bioenergy appear to be consistent. Decomposing EC into SEC and PEC presents a very interesting result. The gross scale inefficiency (SEC) of solar energy significantly impacts its productivity. The competing technical change of solar energy is a result of its technological advancement. All RE alternatives appear to have a relatively equal PEC in 2016 compared to 2012. Productivity analysis of geothermal energy could not be performed due to lack of data across other periods.

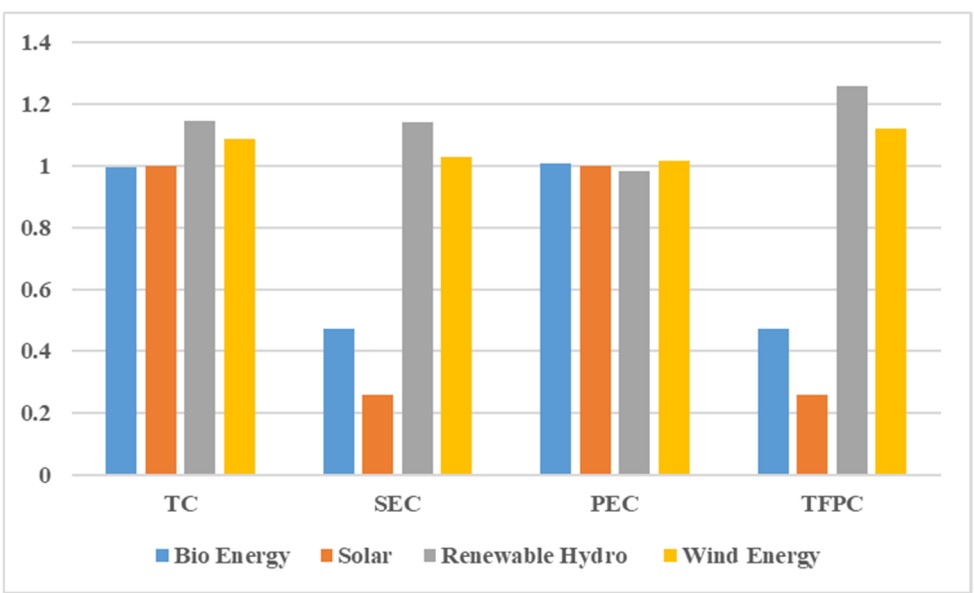

**Figure 4.** Average global Malmquist indices (2012–2016). TC (technical change), SEC (scale efficiency change), PEC (pure efficiency change), TFPC (total factor productivity change).

**Table 2.** Global Malmquist indices by year and country.

| | | TC | SEC | PEC | TFPC |
|---|---|---|---|---|---|
| | **2012–2014** | | | | |
| Bioenergy | Mexico | 1 | 0.31 | 1 | 0.31 |
| | Sweden | 1.05 | 0.4 | 0.95 | 0.4 |
| | UK | 1 | 0.75 | 1 | 0.75 |
| Solar | Chile | 1.18 | 1.06 | 1.06 | 1.151 |
| | Mexico | 1 | 1.47 | 1 | 1.47 |
| Wind Energy | Austria | 1 | 0.94 | 1 | 0.94 |
| | Germany | 1 | 1.02 | 1 | 1.02 |
| | Poland | 1.12 | 1.03 | 0.95 | 1.1 |
| | **2014–2016** | | | | |
| Bioenergy | Italy | 1 | 0.59 | 1 | 0.59 |
| | Sweden | 1.08 | 2.15 | 1 | 2.31 |
| | UK | 1 | 1.05 | 1 | 1.05 |
| Solar | Chile | 1.15 | 1.15 | 0.96 | 1.29 |
| | Mexico | 1 | 0.2 | 1 | 0.2 |
| | Turkey | 1.05 | 0.03 | 0.89 | 0.03 |

**Table 2.** *Cont.*

|  |  | 2014–2016 | | | |
|---|---|---|---|---|---|
|  |  | **TC** | **SEC** | **PEC** | **TFPC** |
| Wind Energy | Austria | 1.01 | 0.62 | 0.99 | 0.62 |
|  | Netherlands | 1.12 | 1.37 | 0.91 | 1.29 |
|  | Poland | 1.12 | 0.43 | 1.01 | 0.49 |
|  | Sweden | 1.01 | 1.32 | 1.09 | 1.09 |
|  | Turkey | 0.96 | 1.45 | 1.09 | 1.52 |
|  | UK | 1 | 1.61 | 1 | 1.61 |
|  |  | 2012–2016 | | | |
| Bioenergy | Finland | 0.99 | 0.16 | 1.02 | 0.16 |
|  | Sweden | 1 | 0.59 | 1 | 0.59 |
| Renewable Hydro | Colombia | 1 | 0.26 | 1 | 0.26 |
| Solar | Chile | 1.21 | 1.56 | 1.11 | 1.25 |
|  | Colombia | 1.08 | 0.01 | 0.86 | 0.01 |
| Wind Energy | Austria | 1.09 | 1.03 | 1 | 1.12 |
|  | Belgium | 1.21 | 0.36 | 0.97 | 0.43 |
|  | Poland | 0.96 | 1.38 | 1.08 | 1.48 |

## 4. Conclusions

This study examined the dual efficiency of RE alternatives considering energy dimension, economic dimension, environmental dimension, and social dimension in selected OECD countries for 2012, 2014, and 2016. The study does not only provide analysis of individual RE alternatives over time, it also provides a comparison with other alternatives for more informed decision making. The analysis has two folds. First, which is the most efficient and productive RE alternative in the selected OECD countries? Second, which RE alternative is best for a particular county? To analyze efficiency, a VRS DEA model was utilized as well as the MPI for productivity analysis. First, the result presented for efficiency shows that all RE alternatives were improving in efficiency across the evaluated period. However, bioenergy appears to be the most efficient due to its maximum average efficiency score. Renewable hydro and wind energy shows significant potential as well. Countries' performance in the RE alternatives is non-monolithic. Countries performed differently with regards to the RE alternatives. Therefore, the countries should enhance performance in the RE alternative where they perform better. For example, bioenergy appears to be the most efficient and productive. However, Italy does better in renewable hydro compared to bioenergy. Similarly, Turkey does well in renewable hydro compared to solar and wind energy. Furthermore, prudent economic policies toward RE and strategic investment are required to improve efficiency. Factors such as installed capacity are covered under the input (investment capital). Weather conditions are exogenous factors that are beyond the control of the energy systems and should be considered by individual countries. Therefore, weather conditions and resource availability are factors that countries should carefully analyze when deciding on the RE alternative to pursue in order to achieve RE efficiency.

The present study makes several contributions to the RE efficiency literature. First, it outlines the efficiency dimension of RE systems and considers indicators that adequately represent the dimension in defining efficiency and productivity of RE. A composite and comprehensive indicator was also introduced to adequately account for the complexity posed by the environmental dimension of the RE system. Second, the DEA analysis enhances our understanding of the relative efficiency of RE alternatives, leading to the conclusion that RE efficiency is not monolithic across countries. For instance, Turkey is

efficient in renewable hydro energy and is inefficient in wind energy, solar energy, and bioenergy. Therefore, Turkey should enhance other RE alternatives if they are to operate a mixed RE system. Chile is efficient in bioenergy and inefficient in solar energy, wind energy, and geothermal energy. The UK and Sweden had significant improvement in bioenergy and wind energy in 2016 compared to other periods.

Several noteworthy contributions for policy makers are also provided. First, information context is provided for policy makers aiming to improve efficiency across all RE dimensions. Second, policy makers should understand that RE efficiency tends to be individualistic according to the countries' resources potential and not a generic performance. Exogenous factors should also be considered. Lastly, the findings of this study provide an incentive for policy makers to pursue further development of their efficient RE technologies following the significant growth in efficiency across all RE alternatives.

This study also has a couple of limitations. First, the number of sample countries is not large enough to generalize the findings; however, statistical measures were employed to limit any effects on the results despite them satisfying DEA efficiency evaluation criteria. Data availability was a major constraint in the analysis. Perhaps when countries are fully committed to transition into complete RE systems, efforts toward data availability will improve. Second, the study did not consider the economic dimension as an output due to data restriction. Share of RE contribution to economic measures such as GDP could be considered. Future studies should aim to include a similar output. In addition, future studies may also consider a weight-restricted DEA model. However, concrete and evidence-based weight selection should be made before allocating weights to the selected variables. PROMETHEE method or analytical hierarchy process (AHP) are interesting multi-criteria decision techniques to rank and allocate weights to the decision variables.

**Author Contributions:** Conceptualization, S.E.K., M.D.I., and S.D.; methodology, S.E.K. and M.D.I. software, S.D.; validation, S.E.K., M.D.I., and S.D.; formal analysis, S.E.K. and M.D.I.; resources, S.E.K.; data curation, M.D.I.; writing—original draft preparation, S.E.K. and M.D.I.; writing—review and editing, S.E.K. and M.D.I.; visualization, M.D.I.; supervision, S.D.; project administration, S.E.K. All authors have read and agreed to the published version of the manuscript.

**Funding:** This research received no external funding.

**Institutional Review Board Statement:** Not applicable.

**Informed Consent Statement:** Not applicable.

**Data Availability Statement:** The data presented in this study are openly available in Mendeley Data at doi:10.17632/chpjt7wfz7.1.

**Conflicts of Interest:** The authors declare no conflict of interest.

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
