# Peer review of "Dual Efficiency and Productivity Analysis of Renewable Energy Alternatives of OECD Countries"

_sustainability, doi:10.3390/su13137401_

Round 1
Reviewer 1 Report
The paper examines the dual efficiency of several Renewable Energy Sources (RES) for selected OECD countries through an integrated model with energy, economic, environmental, and social dimensions. The main goal was to analyze which RES is more dual efficient and productive. This paper should present an enhanced previous version of the paper that was rejected from publication. Compared to the first version of the paper the introduction chapter is enhanced a lot and the used term dual efficiency is described. However, the paper is still not well organized and has many shortcomings:
- Chapter 2.1 seems to be more suitable in the introduction chapter.
- Row 197: reference should be added behind the “SDG13, 196 SDG14, and SDG15”.
- Row 147: remove one dot.
- Row 149: add a dot at the end of the row.
- Row 157: dot is missing behind the reference.
- Row 163: capitalize each word - Environmental Performance Index (EPI)
- Rows 218-220: “It is developed using 32 performance indicators across 11 issue categories under two major issues-Environmental health and ecosystem vitality.” – please describe EPI in detail in the appendix.
- The main techniques used in evaluation (described in sections 2.3 and 2.4) are not clearly described. These two chapters should be described more in detail, and, if appropriate, an example should be given on how to use them. Large parts of the sections belong to the introduction chapter. The paper isn’t well organized.
- Row 234-235: capitalize each word - Malmquist Productivity Index (MPI)
- Row 243: capitalize each word - Technical Efficiency Change (TEC) or Efficiency Change (EC)
- Row 244: capitalize each word - Frontier Change (FC) or Technical Change (TC)
- Input data are not shown so nobody can check calculated results. This means that simulations cannot be repeated.
- The obtained results and conclusions are not clear. Each country has its specifics so it seems that comparison on efficiencies could not be given. Only efficiencies over a period of a selected country can be compared but there should be several factors considered like installed power, weather conditions of each year, etc., which is not clear from the paper whether authors considered that.
- The weight distribution of the variables is not shown.
Author Response
The paper examines the dual efficiency of several Renewable Energy Sources (RES) for selected OECD countries through an integrated model with energy, economic, environmental, and social dimensions. The main goal was to analyze which RES is more dual efficient and productive. This paper should present an enhanced previous version of the paper that was rejected from publication. Compared to the first version of the paper the introduction chapter is enhanced a lot and the used term dual efficiency is described. However, the paper is still not well organized and has many shortcomings:
- Chapter 2.1 seems to be more suitable in the introduction chapter.
Response:
Thank you for your valuable observation. We have moved section 2.1 to the introduction and integrate it accordingly.
- Row 197: reference should be added behind the “SDG13, 196 SDG14, and SDG15”.
Response:
Thank you for your comments. We have added appropriate references.
- Row 147: remove one dot.
Response:
Thank you for your comments. Extra punctuation has been removed.
- Row 149: add a dot at the end of the row.
Response:
Thank you for your comment. Punctuation has been added
- Row 157: dot is missing behind the reference.
Response:
Thank you for your comment. Punctuation has been added
- Row 163: capitalize each word - Environmental Performance Index (EPI)
Response:
Thank you for your comment. It has been modified upper case.
- Rows 218-220: “It is developed using 32 performance indicators across 11 issue categories under two major issues-Environmental health and ecosystem vitality.” – please describe EPI in detail in the appendix.
Response:
Thank you for your comment. Figure 1 has been added to describe the composition of EPI.
- The main techniques used in evaluation (described in sections 2.3 and 2.4) are not clearly described. These two chapters should be described more in detail, and, if appropriate, an example should be given on how to use them. Large parts of the sections belong to the introduction chapter. The paper isn’t well organized.
Response:
Thank you for your valuable observations. The mentioned sections has been improved with an illustrative example and a Figure for the technique.
- Row 234-235: capitalize each word - Malmquist Productivity Index (MPI)
Response:
Thank you for your comment. It has been modified to upper case.
- Row 243: capitalize each word - Technical Efficiency Change (TEC) or Efficiency Change (EC)
Response:
Thank you for your comment. It has been modified to upper case
- Row 244: capitalize each word - Frontier Change (FC) or Technical Change (TC)
Response:
Thank you for your comment. It has been modified to upper case
- Input data are not shown so nobody can check calculated results. This means that simulations cannot be repeated.
Response:
The data presented in this study are openly available in [Mendeley Data] at [DOI: 10.17632/chpjt7wfz7.1].
- The obtained results and conclusions are not clear. Each country has its specifics so it seems that comparison on efficiencies could not be given. Only efficiencies over a period of a selected country can be compared but there should be several factors considered like installed power, weather conditions of each year, etc., which is not clear from the paper whether authors considered that.
Response:
Thank you for your valuable comment. The results and conclusions have been improved. The comments have been clarified at appropriate sections within the manuscript as follows:
“Investment is made in expanding installed capacity and technologies required for RE to usable forms.”
“Efficiency is a universal measurement that can be applied to energy alternatives in this case, RE. The efficiency context and grounds for comparison has to be consistent across all RE alternatives. An array of indicators to define the situation and objective of the analysis needs to be made [9]. In this study, the dual efficiency context and grounds for comparison are consistent across all RE alternatives. Therefore, relative comparison of RE alternatives in different periods across various countries is feasible.”
“For example, Bioenergy appears to be the most efficient and productive. However, Italy does better in Renewable hydro compared to bioenergy. Similarly, Turkey does well in Renewable hydro compared to solar and wind energy. Furthermore, prudent economic policies towards RE and strategic investment are required to improve efficiency. Factors such as installed capacity are covered under the input (Investment capital). Weather conditions are exogenous factors that are beyond the control of the energy systems, and should be considered by individual countries. Therefore, weather conditions and resource availability are factors that countries should carefully analyze when deciding on the RE alternative to pursue in order to achieve RE efficiency”
“Second, policy makers should understand that RE efficiency tend to be individualistic according to the countries resources potential and not a generic performance. Exogenous factors should also be considered.”
- The average weight distribution of the variables is not shown.
Response:
Thank you for your comment. The average weight distribution has been mentioned in the manuscript.
“The average weight distribution (Capital investment= 3.67, Electricity generation from respective RE sources= 0.056, EPI=1.13, Access to clean fuels and technologies= 0.421)”
Reviewer 2 Report
Authors present an article untitled :"Dual efficiency and productivity analysis of renewable energy alternatives of OECD Countries".
The innovation and the original contribution are not enhanced.
OECD is composed by 37 countries but only half are present in this study, what was the metric for such chosen ones?
We are in year 2021, but this study makes a comparison
between years 2012, 2014 and 2016? Why? Doesnt exist data for the rest of the years?
Authors presents 63 references! Almost a Review Paper, withot the review part...
In section 3, in line 269 "Renewable hydro and geothermal energy were
excluded for 2014 and 2016 respectively due insufficient data of those RE alternatives at those periods. This does not affect the stability of the model or reliability of the results as thePPS function remains the same.".
How is this possible? without sufficient data, the results remain reliable??
This paper need significant improvement
Author Response
Authors present an article untitled: "Dual efficiency and productivity analysis of renewable energy alternatives of OECD Countries".
OECD is composed by 37 countries but only half are present in this study, what was the metric for such chosen ones?
Response:
Thank you for your comment.
Despite growing interest in RE, only few countries pursue the possibility of complete transition to RE, thus the limited data availability for RE when to compared to traditional energy data. However, data limitation does not hinder analysis, provided the utilized models are stable for the results to be useful.
We are in year 2021, but this study makes a comparison between years 2012, 2014 and 2016? Why? Doesn’t exist data for the rest of the years?
Response:
Thank you for your comment.
Efficiency analysis and energy efficiency in particular is not time dependent. Comparison can be made and insight gained about the performance at any point to enhance future improvement.
Authors presents 63 references! Almost a Review Paper, without the review part...
Response:
Thank you for comment. The authors considered the manuscript volume. Literature review section was not included since there are adequate literature review articles on the topic which are summarized and cited in the study. The number of references only seek to give proper citations in the study when necessary.
In section 3, in line 269 "Renewable hydro and geothermal energy were
excluded for 2014 and 2016 respectively due insufficient data of those RE alternatives at those periods. This does not affect the stability of the model or reliability of the results as the PPS function remains the same." How is this possible? without sufficient data, the results remain reliable??
Response:
Thank you for your comment.
If a model is stable, the results are reliable. The limited data does not hinder analysis, provided the model adequately handles the available data.
The Authors made the following clarifications in the manuscript.
Line 292 – 296 “To ensure empirical stability of DEA models, the number of evaluated units “n” must satisfy the criteria: [64]. Considering the one input three output, number of units should be greater than or equal to twelve. The study evaluates sixty seven units, therefore the model is stable, and results are reliable”
Round 2
Reviewer 1 Report
The paper examines the dual efficiency of several Renewable Energy Sources (RES) for selected OECD countries through an integrated model with energy, economic, environmental, and social dimensions. The main goal was to analyze which RES is more dual efficient and productive.
The article is a revised version that has been significantly improved over the previous version that was not accepted for publication. All my recommendations have been accepted and the article now looks much better and is more understandable. My suggestion is to accept it in its present form.
Reviewer 2 Report
Authors have addressed reviewers suggestions and improved this document, thus, is ready for publish
This manuscript is a resubmission of an earlier submission. The following is a list of the peer review reports and author responses from that submission.
Round 1
Reviewer 1 Report
Authors present an article untitled :"Dual efficiency and productivity analysis of renewable energy alternatives of OECD Countries".
First question: What is dual efficiency?
The introduction consists of a set of loose sentences, but they do not form a coherent text. The state of art is not complete.
The innovation and the original contribution are not enhanced.
OECD is composed by 37 countries but only half are present in this study, what was the metric for such chosen ones?
We are in year 2021, but this study makes a comparison between years 2012, 2014 and 2016? Why? Does not exist data for the rest of the years?
Authors presents 75 references! Almost a Review Paper, withot the review part...
In section 3, line 299 " Due to data availability, the renewable energy alternatives evaluated were bioenergy, renewable-hydro, solar energy, wind energy, and geothermal energy for the available OECD countries", but in line 302 "Renewable hydro and geothermal energy were excluded for 2014 and 2016 respectively due imbalanced data set".
What study is this?? without sufficient data??
Reviewer 2 Report
The paper examines the dual efficiency of several Renewable Energy Sources (RES) for selected OECD countries through an integrated model with energy, economic, environmental, and social dimensions. The main goal was to analyze which RES is more dual efficient and productive.
The paper is not well organized and has several shortcomings:
- The introduction part is too long. It seems that chapter 1.1. is not necessary since it gives general information regarding RES.
- The main techniques used in evaluation (described in sections 2.3 and 2.3) are not clearly described. These two chapters should be described more in detail, and, if appropriate, an example should be given on how to use them.
- Input data are not shown so nobody can check calculated results. This means that simulations cannot be repeated.
- The obtained results and conclusions are not clear. Each country has its specifics so it seems that a comparison on efficiencies could not be given. Only efficiencies over a period of a selected country can be compared but there should be several factors taken into account like installed power, weather conditions of each year, etc., which is not clear from the paper whether authors considered that.
- The paper has several punctuation mistakes and should be double checked before submitting.
- For Renewable Energy Sources suggestion is to use widely accepted RES abbreviation instead RE.